

# Stress responses of the oil-producing green microalga *Botryococcus braunii* Race B

Ivette Cornejo-Corona[1], Hem R. Thapa[2], Daniel R. Browne[2], Timothy P. Devarenne[2] and Edmundo Lozoya-Gloria[1]

[1] Genetic Engineering, Centro de Investigación y de Estudios Avanzados del Instituto Politécnico Nacional, Unidad Irapuato, Irapuato, Guanajuato, Mexico

[2] Department of Biochemistry and Biophysics, Texas A&M University, College Station, TX, United States

## ABSTRACT

Plants react to biotic and abiotic stresses with a variety of responses including the production of reactive oxygen species (ROS), which may result in programmed cell death (PCD). The mechanisms underlying ROS production and PCD have not been well studied in microalgae. Here, we analyzed ROS accumulation, biomass accumulation, and hydrocarbon production in the colony-forming green microalga *Botryococcus braunii* in response to several stress inducers such as NaCl, NaHCO$_3$, salicylic acid (SA), methyl jasmonate, and acetic acid. We also identified and cloned a single cDNA for the *B. braunii* ortholog of the *Arabidopsis* gene *defender against cell death 1* (*DAD1*), a gene that is directly involved in PCD regulation. The function of *B. braunii DAD1* was assessed by a complementation assay of the yeast knockout line of the *DAD1* ortholog, oligosaccharyl transferase 2. Additionally, we found that *DAD1* transcription was induced in response to SA at short times. These results suggest that *B. braunii* responds to stresses by mechanisms similar to those in land plants and other organisms.

## INTRODUCTION

Photosynthetic organisms support life on Earth by emitting oxygen as a byproduct of the photosynthetic process, and as a result these organisms face constant photo-oxidative stress (*Ledford & Niyogi, 2005*). Most of the reactions involved in the capture of light energy are related to the production and control of reactive oxygen species (ROS) (*Mittler et al., 2004*; *Hebelstrup & Møller, 2015*). ROS generally exist as different forms of oxygen depending on interactions with an added electron or proton. ROS may be present as superoxide radical, hydrogen peroxide, or a hydroxyl radical (*Mallick & Mohn, 2000*). The equilibrium between light harvesting and energy production must be carefully controlled, otherwise the produced ROS may result in loss of protein function, deterioration of membrane integrity, and cell death (*Ledford & Niyogi, 2005*). Non-damaging levels of ROS may prepare cells to deal with higher ROS levels in order to survive this oxidative stress condition (*Ledford & Niyogi, 2005*). Mechanisms to deal with ROS include non-enzymatic ROS scavenging

Corresponding author
Edmundo Lozoya-Gloria,
edmundo.lozoya@cinvestav.mx

through the production of antioxidant compounds like carotenoids (*Apel & Hirt, 2004*). However, excessive ROS production may induce the process known as programmed cell death (PCD), which is an active and genetically controlled process initiated during normal development, and utilized to isolate and remove damaged tissues to ensure survival of the organism (*Petrov et al., 2015*). Different kinds of biotic and abiotic stresses may also trigger ROS production leading to PCD (*Torres, Jones & Dangl, 2006*).

Studies on PCD in land plants have received substantial attention, whereas PCD in algae has not been well studied. In the unicellular freshwater green alga *Micrasterias denticulate* some biochemical and physiological modifications characteristic of PCD were shown to be induced by high salt (NaCl or KCl) similar to what has been seen for higher plants (*Affenzeller et al., 2009*). After long exposure (24 h) to these stresses autophagy of organelles was detected, which is a particular type of PCD (*Affenzeller et al., 2009*). In *Chlamydomonas reinhardtii*, acetic acid, NaCl, and $Na_2CO_3$ induced PCD and the release of volatile compounds (*Zuo et al., 2015*). Apoptosis is another form of PCD that is accompanied by specific changes in morphology, such as increased chromatin condensation, nuclear degradation, and DNA fragmentation (*Van Doorn & Woltering, 2005*). These changes were observed during PCD in the unicellular chlorophyte *Dunaliella tertiolecta* and were accompanied by increased expression of caspases, which are cell death-associated proteases (*Segovia et al., 2003*). It is known that after oxidative stress, cells somehow sense ROS production and specific signal transduction processes activate transcription factors which trigger gene expression of caspases and other degrading proteins to bring about PCD (*Rantong & Gunawardena, 2015*). Expression of PCD-related genes was studied in *C. reinhardtii* after PCD induction by UV-C treatment. This study showed an increase of protein levels for apoptosis protease activating factor-1 (AFA1), and down-regulation of expression for the anti-apoptotic gene *defender against death* (*DAD1*) (*Moharikar, D'Souza & Rao, 2007*). In the regulatory framework of PCD, *DAD1* has been recognized as a key inhibitor of PCD. The highly conserved *DAD1* gene has been identified in distantly related organisms such as human, mouse, rat, chicken, *Xenopus*, *C. elegans*, yeast, and plants (*Wang et al., 1997*; *Van der Kop et al., 2003*). The *DAD1* gene was initially isolated from a temperature-sensitive hamster mutant cell line, and the encoded protein can inhibit the development of PCD in *C. elegans* (*Nakashima et al., 1993*). The DAD1 protein was identified as part of the oligosaccharyl transferase (OST) complex and its activity is related to N-linked glycosylation of the apoptotic machinery (*Nishimoto, 1997*; *Knauer & Lehle, 1999*). Thus, DAD1 is evolutionarily conserved as a universal negative regulator of PCD (*Makishima et al., 1997*).

The green microalga *Botryococcus braunii* is a colonial alga with individual cells of the colony held together by an extracellular matrix (ECM), and produces high levels of liquid hydrocarbons that are mainly stored in the ECM (*Banerjee et al., 2002*). Three races of *B. braunii* have been described, which are differentiated by the type of liquid hydrocarbon produced. Race B hydrocarbons are polymethylated, unsaturated triterpenes between 30–37 carbons named botryococcenes, race L produces a tetraterpene named lycopadiene, and race A produces fatty acid-derived alkadienes and alkatrienes of 25–31 carbons (*Banerjee et al., 2002*). Although ROS were not directly assayed, a study placing *B. braunii* Races B

and L under stress conditions such as nitrogen deficiency or high light intensity showed accumulation of carotenoids that could act as antioxidants capable of quenching ROS that may be produced under these conditions (*Ambati, Ravi & Aswathanarayana, 2010*). During the linear phase of the *B. braunii* growth cycle, antioxidants/pigments such as lutein were produced, whereas the antioxidants/pigments echinenone and canthaxanthin were produced mainly in the stationary phase in response to depletion of nitrogen (*Grung, Metzger & Liaaen-Jensen, 1989*). This data suggests that ROS could be produced in *B. braunii* under certain stress conditions (*Grung et al., 1994*). The ROS protective antioxidant properties of the acetone extracted carotenoids from *B. braunii* were demonstrated *in vitro* using systems such as the 2,2-diphenyl-1-picrylhydrazyl (DPPH) antioxidant assay in human low-density lipoprotein and rat tissues (*Rao et al., 2006*; *Rao et al., 2013*). Even though the production of these antioxidant compounds in *B. braunii* has been known for some time, there has not been a report about the production of ROS or the induction of PCD in this alga. In this work, we present results showing ROS production in *B. braunii* after treatment with several common stress inducers using a staining procedure adapted from mammalian systems. We also demonstrate the existence of the *DAD1* gene in *B. braunii*, assess its function by complementation of the yeast knockout line of the *DAD1* homologue (oligosaccharyl transferase 2, *OST2*), and show induction of *DAD1* gene expression after stress treatments.

## MATERIALS AND METHODS

### Algal culturing

Batch cultures of *B. braunii* Race B, Showa (Berkeley) strain (*Nonomura, 1988*) were grown in 500 mL homemade bioreactors by inoculating 40 mL of a two-week-old culture into 400 mL fresh Chu-13 media (*Grung, Metzger & Liaaen-Jensen, 1989*). The algal cultures were aerated with filter-sterilized ambient air and grown at 25 °C under cool-white fluorescent illumination at photosynthetically active radiation of 50 $\mu$mol photons m$^{-2}$s$^{-1}$ with 59 watt lamps (Philip F96T8 59W/850 Single Pin ALTO Plus). Cultures were grown for two weeks to obtain green active state colonies prior to analysis in accordance with previous reports (*Brown, Knights & Conway, 1969*). Algal cells of two-week-old cultures were collected by filtration using Millipore Nylon filters (10 $\mu$m pore size) and the dry weight (DW) biomass was calculated after filtration in pre-weighed 0.45 $\mu$m Millipore Nylon membranes followed by drying in an oven at 65 °C for 24 h.

### Induction of stress conditions

Before stress induction, cells from two-week-old cultures were washed by filtration through a 10 $\mu$m nylon net followed by suspension of cells in 100 mL of sterile deionized water. For stress induction 400 mL of fresh media was inoculated with 10 mL of water-suspended cells (0.6 g DW L$^{-1}$), cultures grown for 72 h, and independent cultures treated with 50, 100, and 200 mM NaCl (KARAL); 120, 240, and 360 mM NaHCO$_3$ (KARAL); 1.5, 3.0, and 6.0 mM salicylic acid (SA; Sigma-Aldrich); 10, 20, and 30 $\mu$M methyl jasmonate (MeJA; Sigma-Aldrich) and 1.09, 2.18, and 4.33 mM acetic acid (KARAL). Samples were taken at

10, 30, 60, and 120 min after treatment and the cultures were either used for ROS detection by fluorescent staining or the cells were harvested by filtration, frozen in liquid $N_2$, and stored at $-80\,^\circ$C for future use.

### Staining procedure for ROS detection in *B. braunii* colonies

We optimize the method to detect ROS in *B. braunii* colonies by modifying the procedure for CellROX Green reagent (Invitrogen) recommended by the supplier. Briefly, 2 μL of 5 mM CellROX Green was added to 100 μL of algal culture followed by incubation at 36 °C and shaking at 120 rpm for 30 min in the dark. The cells were then washed twice for 5 min each time at room temperature with 1× PBS, 0.1% Triton X-100, and fluorescence was observed using a Zeiss Axio Lab.A1 microscope with a ICc3 Rev.3 digital camera under the control of ZEN lite 2011 Software. An LED module at 470 nm was used for excitation and a filter set of 38 Endow GFP shift free (E) EX BP 470/40, BS FT 495, EM BP 525/50 was used for emission detection. At least one hundred colonies were evaluated for each treatment. The samples were observed under white light to locate colonies for evaluation and then the microscope was switched to fluorescence to identify and count the number of fluorescent colonies. Due to the colony organization of *B. braunii* it was sometimes difficult to count individual cells in a selected sample without moving the field under the microscope. So, once the field was fixed under fluorescent light colonies were considered ROS positive if more than 90% of the cells in the field had fluorescent nuclei.

The percentage of ROS positive colonies was determined according to the equation:

$$(FC/TC) * 100 = \%PC$$

where FC = number of colonies with fluorescence, TC = total number of observed colonies, and PC = percent of ROS positive colonies.

### *B. braunii* hydrocarbon production

Samples were collected by filtration as described above and DW was measured. The extracellular hydrocarbons were extracted three times by the addition of 50 mL of *n*-hexane and incubation for 60 min with continuous stirring at 1,800 rpm using a magnetic stir bar. All *n*-hexane extracts were mixed and evaporated in a rotary evaporator (Büchi). Hydrocarbon amounts were calculated gravimetrically and expressed as a percentage of biomass DW. The average hydrocarbon productivity was evaluated using the data from samples taken every 24 h for seven days after treatments. For botryococcene purification, a gravity-fed silica gel 60 Å (Sigma-Aldrich) column with *n*-hexane as mobile phase was used to separate the pigments from hydrocarbons in the crude extracts. Positive identification of botryococcenes in the hydrocarbon fraction was confirmed by GC-MS (Bruker 436-GC-SCION SQ Premium) using a WCOT BR-5 ms fused silica capillary column (60 m × 0.25 mm, film thickness: 0.25 μm) in electron ionization mode (70 eV) mode. A split injection mode was used with a split ratio of 1:200. Initial oven temperature was 50 °C for 1 min, then increased to 220 °C at a rate of 10 °C min$^{-1}$, and then ramped to 260 °C at a rate of 2 °C min$^{-1}$ and held for 20 min. Total analysis time was 58 min. Helium was used

**Table 1** Most similar defensive and/or stress genes in the *B. braunii* transcriptome in comparison to those from *Arabidopsis*.

| Query | Symbol | Name | Stress | % ID | Reference |
|---|---|---|---|---|---|
| AT1G32210 | *DAD1, ATDAD1* | Defender against death (DAD family) protein | Anti-apoptotic and PCD regulation | 68.7 | *Makishima et al. (1997)* |
| AT2G42680 | *ATMBF1A, MBF1A* | Multiprotein-bridging factor | Abiotic stress tolerance | 64.1 | *Suzuki et al. (2005)* |
| AT5G23140.1 | *CLPP2, NCLPP7* | Nuclear-encoded CLP protease P7 | Proteolytic enzyme activated in response to oxidative stress | 58.5 | *Adam et al. (2001)* |
| AT2G31660.1 | *ARM, RAN* | ARM repeat superfamily protein | SA response | 51.2 | *Mudgil et al. (2004)* |
| AT3G11050.1 | *ATFER2, FER2* | Ferritin 2 | Oxidative stress response, ROS scavenging, pathogen and senescence response | 45.1 | *Murgia et al. (2007)* and *Tarantino et al. (2003)* |
| AT2G14580.1 | *ATPRB1, PRB1* | Basic pathogenesis-related protein 1 | Response to ethylene and MeJA, repression in response to SA | 44.7 | *Mitsuhara et al. (2008)* |
| AT3G49660.1 | *WD40* | Transducing/WD40 Repeat-like super family protein | Regulated by abscisic acid (ABA) | 44.2 | *Sánchez et al. (2004)* |
| AT4G32320.1 | *APX6* | Ascorbate peroxidase 6 | Oxidative stress and heat | 43.2 | *Panchuk et al. (2002)* |

as a carrier gas at a flow rate of 2.50 mL min$^{-1}$ and controlled by linear velocity mode. Temperatures of injection port, interface, and ion source were 280 °C, 250 °C, 200 °C, respectively.

## Bioinformatic analysis and selection of genes related to stress conditions

In order to identify genes from *B. braunii* related to stress responses we first identified genes from the *Arabidopsis thaliana* genome annotated as defense or stress related genes. This analysis returned 382 defense related and 611 stress related genes (Table S1). All genes were utilized as BLAST queries against the *C. reinhardtii* v 5.3 genome (https://phytozome.jgi.doe.gov/pz/portal.html) and the resulting sequences were used in a TBLASTN or nucleotide BLASTN search against the *B. braunii* Race B transcriptome database housed at the University of Kentucky (http://lims.ca.uky.edu/niehausblast/blast/blast.html) to identify orthologous sequences in *B. braunii*. Additionally, *C. reinhardtii* genes classified by their functional annotations as defense, stress, or cell death related were used to identify orthologous sequences in *B. braunii* with the Algal Functional Annotation Tool (*Lopez et al., 2011*) in the *B. braunii* Race B transcriptome database housed at the University of California, Los Angeles (http://pathways-pellegrini.mcdb.ucla.edu/botryo1/). The resulting contigs from the last search were analyzed by biological function and similarity between *B. braunii* and *A. thaliana* as shown in Table 1.

*B. braunii* Race B RNA-seq data obtained from the Chappell Lab at the University of Kentucky was used to construct a transcriptome assembly for analysis of gene expression levels. The RNA-seq data consisted of six Illumina Genome Analyzer IIx single-end libraries prepared from RNA extracted over an algae growth cycle at days 0, 3, 7, 14, 21, and 28. The data was assembled into a *de novo* transcriptome using Trinity v2.0.6 compiled on an

Intel x86-64 computer cluster. Expression values for the *B. braunii* orthologs of the genes in Table 1 were determined for transcripts at each time point, measured in fragments per kilobase per million mapped reads (FPKM), using the standard Trinity gene expression analysis pipeline. To identify genes in *B. braunii*, the transcriptome was queried with the selected *A. thaliana* protein sequences using the TBLASTN algorithm in the BLAST+ v2.2.29 package. TBLASTN hits with an *E*-value lower than 1e–5 were selected for further analysis. The expression levels of the transcripts identified in the *B. braunii* transcriptome were analyzed and resulting *B. braunii* nucleotide sequences were used to design specific primers for transcription analysis by RT-PCR.

## Cloning of the *B. braunii DAD1* cDNA

Total RNA was purified (*López-Gómez & Gómez-Lim, 1992*) from algal cells harvested from a 21-day-old culture. Reverse transcription reactions were carried out using SuperScript III Reverse Transcriptase (RT) (Invitrogen) with an oligo-dT primer according to the supplier's instructions. The generated single-strand cDNA was treated with RNAse H (Invitrogen) for 15 min at 37 °C before use as a template for PCR. Reactions for the *DAD1* gene were performed from the start codon to the stop codon using forward primer (5′- ATGGACACTCTTAAGCTAAT-3′, start codon underlined) and reverse primer (5′- TTAACCCATGAAGTTCCACGC-3′, stop codon underlined). The PCR reaction was carried out using 1 μl of single-strand cDNA in a final volume of 25 μl with GoTaq Green Master Mix (Promega). The Pre-denaturation step was at 95 °C for 60 s, followed by 35 cycles of 95 °C for 30 s, 60 °C for 60 s, 70 °C 30 s, and a final elongation step at 70 °C for 5 min. The PCR fragments were directly cloned into the pGEM-T vector (Promega) according to the supplier instructions. Cloned fragments were sequenced using the Big Dye direct sequencing system (Thermo Fisher) with M13 forward and reverse primers using the supplier's recommendations.

## Functional characterization of *B. braunii DAD1* using the *OST2* deletion yeast strain

The *B. braunii DAD1*–cDNA was subcloned into the yeast vector p426GPD (*Mumberg, Mailer & Funk, 1995*) using the *Bam* HI and *Sal* I restriction sites for expression under the control of the GPD promoter. The *Bam* HI and *Sal* I restriction sites were added to *DAD1* by PCR amplification using forward primer (*Bam* HI site underlined) 5′-GAC GGATCCATGGACACTCTTAAGCTAAT-3′ and reverse primer (*Sal* I site underlined) 5′- GAC GTCGACACCCATGAAGTTCCACGCCACATAG-3′. The *ost2* knockout (*Winzeler et al., 1999*) (YSC1021 Open Biosystems; Δ*ost2*; *OST2x* Δ*ost2::KanMX*) in yeast strain BY4743 (*MAT a/α his3Δ1/his3Δ1 leu2Δ0 /leu2Δ0 lys2Δ0/LYS2 MET15/met15Δ0 ura3Δ0 /ura3Δ0*) (4741/4742) was transformed with the *DAD1*/p426GPD plasmid and the *OST2*/p5472 plasmid (MoBY ORF Library, Dharmacon™ GE Healthcare Cat. YSC5432) according to the LiAc/SS carrier DNA/PEG method (*Gietz & Woods, 2002*). The plasmid vector p5472 carries a *URA3* selectable marker and a yeast centromere. The *OST2* ORF was PCR amplified from an average of 900 bp upstream of the start codon to an average of 250 bp downstream of the stop codon using DNA template

isolated from the sequenced S288C strain (*Ho et al., 2009*). We confirmed the *OST2* ORF in the plasmid by the restriction pattern after a digestion with *Hind* III and *Eco* RI (Invitrogen). Also, we performed PCR for the *OST2* sequence using the forward primer 5′-AATTTATCAAAGCTGTTTCATTTGC-3′ and reverse primer 5′-AAAATGATCCTGCTCTCTTGATATG-3′, and for the *KanMX* F G418/kanamycin resistance gene using the forward primer 5′-TGATTTTGATGACGACGAGCGTAAT-3′and reverse primer 5′-CTGCAGCGAGGAGCCGTAAT-3′ according to the supplier recommendations (Open Biosystems).

Wild type yeast (BY4743) or the *ost2* knockout strain transformed with the *DAD1*/p426GPD plasmid or the wild-type *OST2* cDNA (*OST2*/p5472 plasmid) were grown at 28 °C in YPD + G418 (200 mg/mL) (Sigma-Aldrich), SD-URA + G418 (200 mg/mL), and YPD growth medium, respectively. The phenotype of the *ost2* knockout expressing *DAD1* or *OST2* was analyzed in the presence or absence of 1 μg/mL tunicamycin (Sigma). Cell viability was analyzed by single plate-serial dilution spotting (SP-SDS) (*Thomas et al., 2015*) after tunicamycin treatment using a starting $OD_{600} = 0.6$ and 5 μl of dilutions ($10^1$–$10^6$) on the plates, which were incubated at 28 °C for 48 h.

## qRT-PCR of the *B. braunii DAD1* gene

The expression fold change of the target *DAD1* gene in the treated samples of *B. braunii* was normalized to *β-tubulin* gene. The samples collected at each time point were used to obtain the cDNA for the real time PCR. All qRT-PCR assays were performed on a CFX96TM real-time PCR detection system (BIO-RAD), and the visualized with the CFX ManagerTM software (BIO-RAD). Final volumes of 20 μl contained 10 μl of iQ Sybr Green super mix 2× (BIO-RAD), 325 nM of both forward and reverse primers, RNase free water and 1 μl of cDNA template. Triplicates of no-template controls were included for each run. To amplify a 97 bp fragment of *DAD1* gene the forward primer was 5′-GCCAATCCAGCTAACAAGGA-3′ and reverse primer 5′-TTCCACGCCACATAGAACAA-3′ and for a 140 bp fragment of the *B. braunii β-tubulin* gene the forward primer was 5′-TCCGTCCTTGATGTTGTCCG-3′ and reverse primer 5′-TCCGGGTACTCCTCACGAAT-3′.

All reactions started with 180 s of initial denaturation at 95 °C, followed by 40 cycles of denaturation at 95 °C for 15 s, annealing step at 60 °C for 30 s, and elongation at 72 °C for 30 s. Melting curve was from 55 °C to 95 °C with increments of 0.5 °C every 5 s.

Data from these qRT-PCR experiments were exported from CFX manager software to Microsoft Excel and the expression fold change was calculated by $\Delta\Delta C_T = (C_{T,\text{Target}} - C_{T,\beta-\text{Tubulin}})_{\text{Time } x} - (C_{T,\text{Target}} - C_{T,\beta-\text{Tubulin}})_{\text{Time 0}}$, where $x$ is a time point other than zero (*Livak & Schmittgen, 2001*). The expression fold change for each time point was compared to the 0 time to indicate the basal level of gene expression prior to any treatment.

## Statistical analysis

Statistical tests were performed using GraphPad Prism version 6.00 for Mac OS X, GraphPad Software, La Jolla California USA (http://www.graphpad.com). Data was visualized using Microsoft Excel for Mac 2011 version 14.1.0.

## RESULTS

### ROS response in *B. braunii* cells during stress induction

ROS production has been reported in plants and algae after treatment with several common stress inducers such as NaCl (*Rao et al., 2007*; *Yilancioglu et al., 2014*; *Pancha et al., 2015*), NaHCO$_3$ (*Gao et al., 2014*), salicylic acid (SA) (*Dorey et al., 1997*; *Gil et al., 2005*), methyl jasmonate (MeJA) (*Küpper et al., 2009*), and acetic acid (*Zuo et al., 2012a*). We used the same inducers to examine if *B. braunii* cells were able to respond to these stresses in a similar manner. To determine if ROS were being produced in stressed *B. braunii* cells, we used a staining assay to detect the presence of intracellular ROS under the different stress treatments. This assay is based on the dye CellROX Green which is a fluorogenic probe used for measuring oxidative stress in live cells (*Cheloni, Cosio & Slaveykova, 2014*). The dye is weakly fluorescent inside cells in a reduced state but exhibits bright green photostable fluorescence upon oxidation by ROS and subsequent binding to DNA (*Gemelli et al., 2014*; *Stodůlková et al., 2015*). Modifications to the procedure recommended by the supplier were made as described in Material and Methods in order to detect ROS positive *B. braunii* colonies.

Using the described procedure, one hundred *B. braunii* colonies were counted after each stress induction treatment and the number of fluorescent colonies with more than 90% of cells having fluorescent nuclei were recorded and expressed as a percentage of the total number of colonies examined. Figure 1 shows the images of *B. braunii* colonies after 60 min of stress inducer treatment with the optimal inducer concentrations (see below). Control and treated colonies looked normal under white light (Figs.1A– 1C and 1G– 1I). After addition of the CellROX reagent, control cultures showed very few fluorescent nuclei (Fig. 1D), while treatment with 100 mM NaCl (Fig. 1E), 120 mM NaHCO$_3$ (Fig. 1F), 3 mM SA (Fig. 1J), 10 µM MeJA (Fig. 1K), and 4.33 mM acetic acid (Fig. 1L) resulted in colonies with most of the nuclei showing fluorescence indicating ROS positive colonies. At longer incubation times of more than 120 min or treatment with higher concentrations of inducers, specific fluorescence intensity decreased and unspecific fluorescence was present most probably due to carotenoids produced by damaged cells (*Davis et al., 2014*; *Morosinotto & Bassi, 2014*) (Fig. S2).

Figure 2 shows the percentages of positive fluorescent colonies at different times and concentrations of inducers. Non-treated colonies (control) show up to 13% ROS positive colonies during the 120 min time course (Fig. 2A), but each treatment showed higher percentages of ROS positive colonies (Figs. 2B–2F ). In all cases, the 60 min time point showed the highest percentage of ROS positive colonies. These data helped to establish the optimal concentrations of inducers to be used in subsequent assays which were 100 mM NaCl (Fig. 2B), 120 mM NaHCO$_3$ (Fig. 2C), 3 mM SA (Fig. 2D), 10 µM MeJA (Fig. 2E), and 4.33 mM acetic acid (Fig. 2F).

### Hydrocarbon and biomass recovery after stress induction

Using the optimal concentration of stress inducers determined above, the average hydrocarbon and biomass production of the treated *B. braunii* colonies was calculated after analysis every 24 h for 7 days after the treatments. Table 2 shows those averages where the

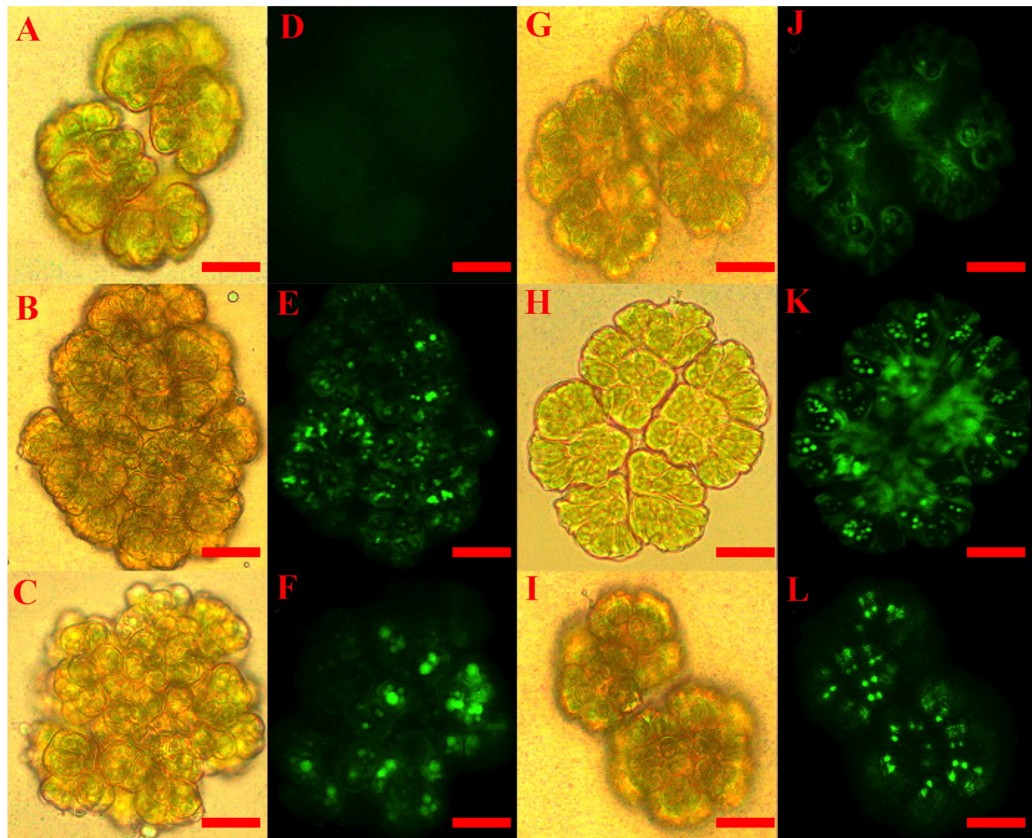

**Figure 1** **Staining of ROS positive colonies of *Botryococcus braunii*.** CellROX dye was used to detect ROS *in vivo* by observing fluorescent *B. braunii* colonies after 60 min treatments. *B. braunii* colonies under white light (A, B, C, G, H and I) and fluorescent excitation (D, E, F, J, K and L). Control without treatment (A, D); and treatments with 100 mM NaCl (B, E), 120 mM NaHCO$_3$ (C, F), 3 mM SA (G, J), 10 $\mu$M MeJA (H, K), and 4.33 mM acetic acid (I, L).

hydrocarbon content of non-treated cells was $31.91 \pm 6.77\%$ with a biomass productivity of $0.047 \pm 0.009$ g DW L$^{-1}$ day$^{-1}$, which is similar to other reports (*Cornejo-Corona et al., 2015*). No significant differences in hydrocarbon or biomass productivity were observed in treated cells compared to the control except for the acetic acid treatment, which showed increased hydrocarbon production over the control (Table 2).

## Identification of *B. braunii* genes related to stress conditions

With the goal of elucidating the response of *B. braunii* genes in relation to the ROS stress conditions used in this study, several characterized and annotated stress and defense response genes from *A. thaliana* were selected as queries for the identification of orthologs in *B. braunii* (see 'Materials and Methods' for detailed description of how these sequences were identified). These genes were related to abiotic stress tolerance, activation by ROS, elicitation by SA and MeJA, involvement in PCD regulation, relation to pathogen and senescence responses, or relation to abscisic acid and heat responses (Table 1).

To assess expression levels of the *B. braunii* genes in Table 1 a transcriptome was made from RNA extracted from samples taken over several days of the *B. braunii* growth cycle.
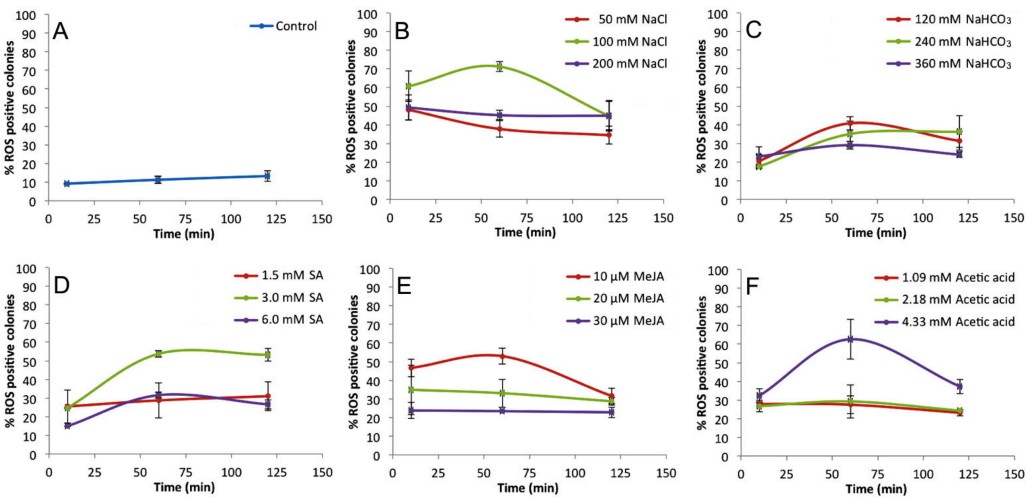

**Figure 2** **Percentage of ROS positive *B. braunii* colonies at different times and concentrations of stress inducers.** Control (A), 100 mM NaCl (B), 120 mM NaHCO$_3$ (C), 3 mM SA (D), 10 μM MeJA (E), and 4.33 mM acetic acid (F). Data represent the mean ± SE of at least three repetitions. Statistical analysis was done by Two-way ANOVA $\alpha = 0.05$, comparisons were against control $P$ value $< 0.0001$****, $P$ value 0.0001 ***, $P$ value 0.0023 **.

**Table 2** **Hydrocarbons and biomass productivity after stress induction in *B. braunii* batch cultures.**[a]

| Treatment | % Hydrocarbon DW | Biomass (g L$^{-1}$ day$^{-1}$) |
| --- | --- | --- |
| Control | 31.91 ± 6.77 | 0.047 ± 0.009 |
| 100 mM NaCl | 31.29 ± 8.46 | 0.049 ± 0.020 |
| 120 mM NaHCO$_3$ | 29.32 ± 4.18 | 0.049 ± 0.012 |
| 10 μM MeJA | 29.60 ± 2.11 | 0.050 ± 0.008 |
| 3 mM SA | 29.04 ± 2.11 | 0.037 ± 0.009 |
| 4.33 mM Acetic acid | 42.56 ± 11.34** | 0.039 ± 0.008 |

**Notes.**
[a] Average of biomass and hydrocarbon productivity was from 12 independent samples evaluated every 24 h by seven days after treatments. Statistical differences are indicated by ANOVA, Tukey HSD.
** $p < 0.002$.

The data was assembled into a de novo transcriptome as described in Materials and Methods and expression values for the selected genes (Table 1) were determined for transcripts at each time point, measured in fragments per kilobase per million mapped reads (FPKM), using the standard Trinity gene expression analysis pipeline. The transcript abundance of several defense and stress related genes in the transcriptome are shown in Fig. 3. Most of the genes (*APX6*, *PRB1*, *ARM*, *CLPP2* and *WD40*) had low and stable transcription along the growth curve, while the *FER2* and *MBF1A* genes were transcribed at higher levels along the growth curve (Fig. 3). Interestingly, the single *DAD1* ortholog identified in the transcriptome had the highest expression during the first 18 days after culture inoculation (Fig. S3 and Table S4).

We decided to focus follow-up studies on the *B. braunii DAD1* gene for several reasons; the interesting *DAD1* transcriptome expression pattern (Fig. 3) during the first 18 days suggest that this gene may be induced during stress conditions. It would be also a useful

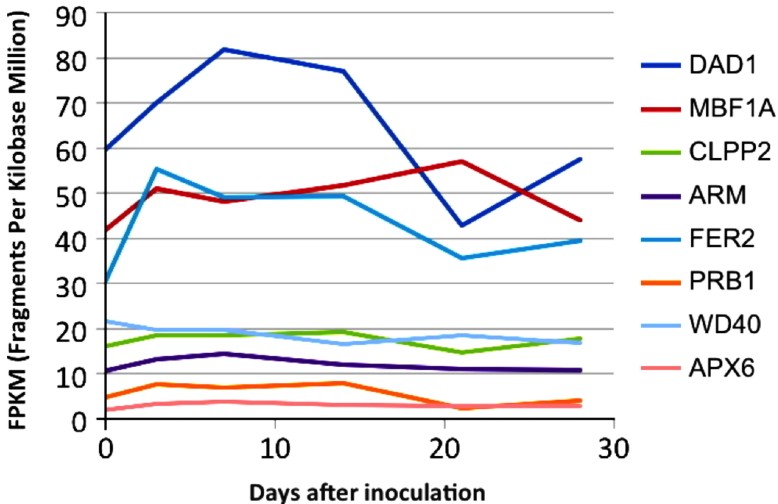

**Figure 3** *In silico* **expression analysis of the selected stress response genes from** ***B. braunii*** **using transcriptome data.** *DAD1*, defender against death; *MBF1A*, multiprotein-bridging factor; *CLPP2*, nuclear-encoded CLP protease P7; *ARM*, ARM repeat superfamily protein; *FER2*, ferritin 2; *PRB1*, basic pathogenesis-related protein 1; *WD40*, transducing/WD40 repeat-like super family protein, and *APX6*, ascorbate peroxidase 6.

target gene to analyze if stress induction by ROS induces gene expression over short time periods as shown by the fluorescent staining studies (Fig. 2). *DAD1* would also be an interesting gene candidate to study for stress responses because the *Arabidopsis DAD1* gene (AT1G32210) is involved in negative regulation of programmed cell death (*Rantong & Gunawardena, 2015*), *DAD1* is expressed during the entire plant life cycle in most if not all the tissues (*Shun-bin et al., 2001*), *DAD1* is conserved among many diverse organisms (*Wang et al., 1997*), and *DAD1* can complement a mammalian apoptosis suppressor mutation (*Gallois et al., 1997*).

## Isolation of the *B. braunii DAD1* cDNA and analysis of the corresponding protein

The *B. braunii DAD1* cDNA was cloned by RT-PCR using *DAD1* specific primers and the isolated cDNA was sequenced showing an open reading frame of 330 nucleotides encoding a predicted protein of 110 amino acids (Fig. 4A). The *B. braunii* DAD1 protein sequence was compared with those from other algae and plants and high similarity between all proteins was seen especially in the transmembrane domain regions (Fig. 4B). The protein also has a highly conserved DAD superfamily domain (pfam02109) spanning residues 7–110 (Fig. 4C).

## Complementation of the yeast *ost2* knockout strain with the *B. braunii DAD1* cDNA

The DAD1 protein from *B. braunii* is 48.3% identical to the Ost2 protein (NP_014746.2), the DAD1 ortholog from *S. cerevisiae*. The *OST2* gene encodes the $\varepsilon$-subunit of the oligosaccharyl transferase complex, which catalyzes the transfer of high mannose oligosaccharides to consensus glycosylation acceptor sites on proteins in the lumen of the

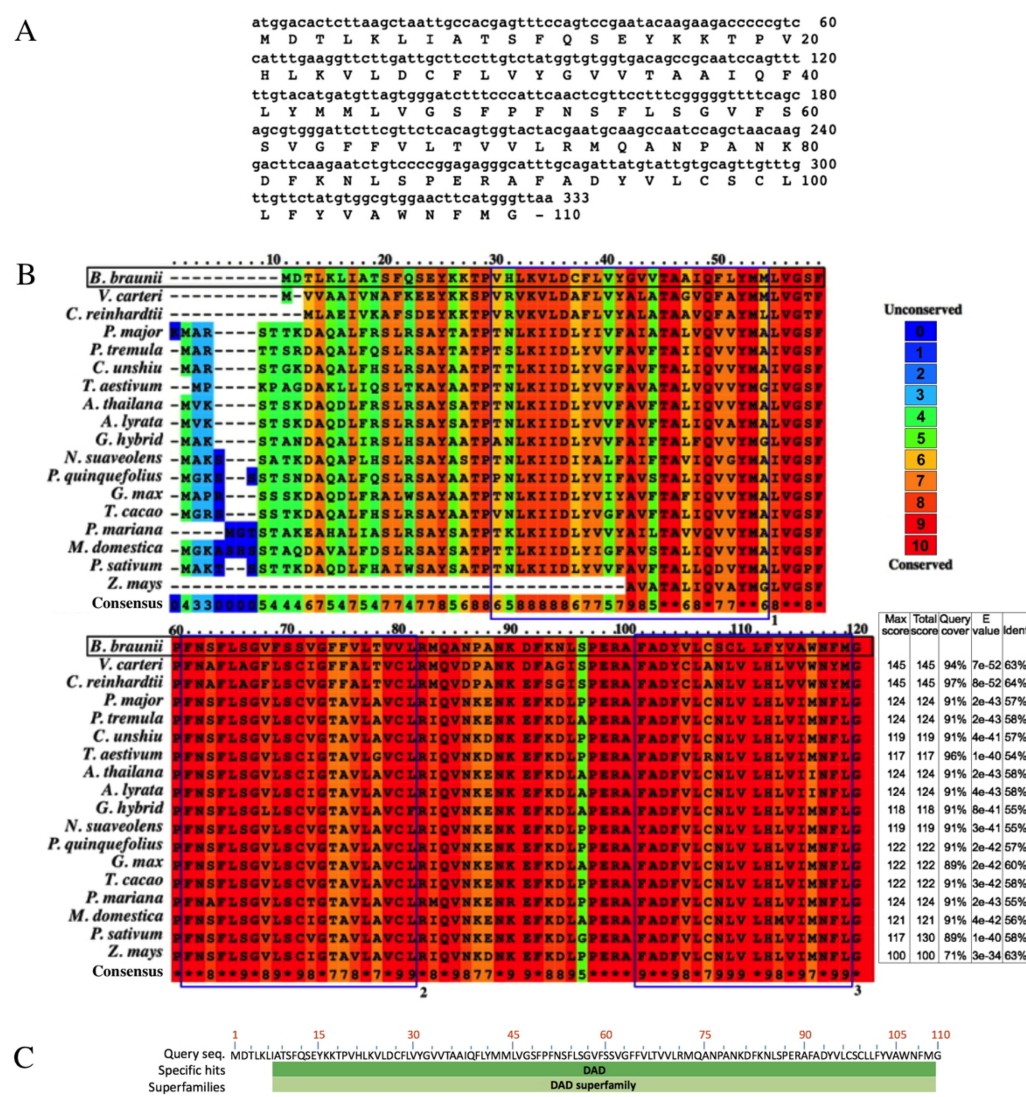

**Figure 4  Nucleotide and amino acid sequences of DAD1, and protein alignment analysis.** Nucleotide and amino acid sequences of the *B. braunii* DAD1 (A), Alignment of DAD1 protein sequences from (B) *Volvox carteri* (XP_002956617), *Chlamydomonas reinhardtii* (XP_001699953), *Plantago major* (AM_156929), *Populus tremula* (AF086839), *Citrus unshiu* (AB011798), *Triticum aestivum* (GU564293), *Arabidopsis thaliana* (NM_102954), *Arabidopsis lyrata* (XP_002893712), *Gladiolus hybrid* (BAD11075), *Nicotiana suaveolens* (AB058922), *Panax quinquefolius* (EU274651), *Glycine max* (NP_001236226), *Theobroma cacao* (EOY25213.1), *Picea mariana* (AF_051247), *Malus domestica* (NP_001280846), *Pisum sativum* (AAC77357), and *Zea mays* (AF055909). Alignment was generated using PRALINE multiple sequence program. The histogram depicts level of conservation for each amino acid. Blue boxes labeled 1, 2, and 3 indicate the putative transmembrane domains. *E*-values and % of identity in comparison with *B. braunii* DAD1 are shown at the end of the sequences. The DAD superfamily domain (pfam02109) map was generated using blastp (C).

rough endoplasmic reticulum (*Zhou et al., 2007*). It was previously reported that genomic disruption of the *OST2* locus was lethal in haploid yeast (*Silberstein et al., 1995*). If *DAD1* of *B. braunii* has a similar function to that of *OST2*, the algal cDNA should complement the *ost2* knockout phenotype. DAD1 function in glycosylation can be analyzed using the antibiotic tunicamycin, which blocks N-linked glycosylation and causes G1 phase cell cycle arrest by inhibition of cell wall biosynthesis in bacteria, yeast, and fungi (*Kuo & Lampen, 1976*; *Hauptmann et al., 2006*; *Sugiura & Takagi, 2006*; *Yang et al., 2009*). Tunicamycin resistance by complementation of the yeast *ost2* knockout has been reported in previous studies using the *A. thaliana DAD1* sequence (*Gallois et al., 1997*) and the *S. cerevisiae OST2* sequence (*Sugiura & Takagi, 2006*).

Considering that the *B. braunii DAD1* cDNA is predicted to encode a complete functional protein it should restore yeast *OST2* function in the *ost2* knockout and allow for growth in the presence of tunicamycin. To test this, the full length *B. braunii DAD1* cDNA was cloned into a yeast expression plasmid and used to transform the yeast *ost2* knockout line. The resulting transformant was tested for growth in the presence of tunicamycin. Wild-type yeast and the *ost2* knockout line are sensitive to tunicamycin treatment equally (*Xiao, Smeekens & Wu, 2016*). As expected, overexpression of *B. braunii* cDNA in wild type or the *ost2* knockout line imparts resistance to tunicamycin (Figs. 5A and 5B).

### qRT-PCR of the *B. braunii DAD1* gene

*DAD1* gene expression was quantitatively analyzed by the $2^{-\Delta\Delta CT}$ method (*Livak & Schmittgen, 2001*; *Zhang, Ruschhaupt & Biczok, 2010*) in order to determine relative changes in *DAD*1 gene expression under different stress treatments over a time course. A relative quantification against $\beta-tubulin$ gene expression in short times were evaluated (Fig. 6). When compared to control condition, *DAD*1 gene expression was similar to that of 100 mM NaCl and 4.33 mM acetic acid treatments. With 120 mM NaHCO$_3$ treatment, *DAD*1 gene expression significantly decreased at 10 min followed by significant increase at 30 min while an opposite gene expression pattern was observed with 3.0 mM salicylic acid treatment which showed higher gene expression at 10 min but decreased at 30 min. Under 10 μM MeJA, *DAD*1 gene expression decreased during the 10 and 30 min. In general, *DAD*1 gene expression was stable in all utilized stress conditions at 60 min (Fig. 6).

## DISCUSSION

### ROS response in *B. braunii* cells during stress induction

Among the stress conditions tested, NaCl was the strongest and fastest ROS inducer (Fig. 2B). All concentrations of NaCl used were statistically higher in ROS accumulation than the control after 10 min of treatment, and the highest induction was seen at 60 min with 100 mM NaCl (Fig. 2B). NaHCO$_3$ was the weakest ROS inducer although the percent of ROS positive colonies was always higher than the control (Fig. 2C).

These results clearly show that *B. braunii* produces ROS very quickly (10 min) after treatment with all stress inducers tested. While the effects of NaCl and NaHCO$_3$ on *B. braunii* have been previously studied, those reports describe effects on growth rate or an increase in carotenoid production, but do not report ROS production (*Li & Qin, 2005*;

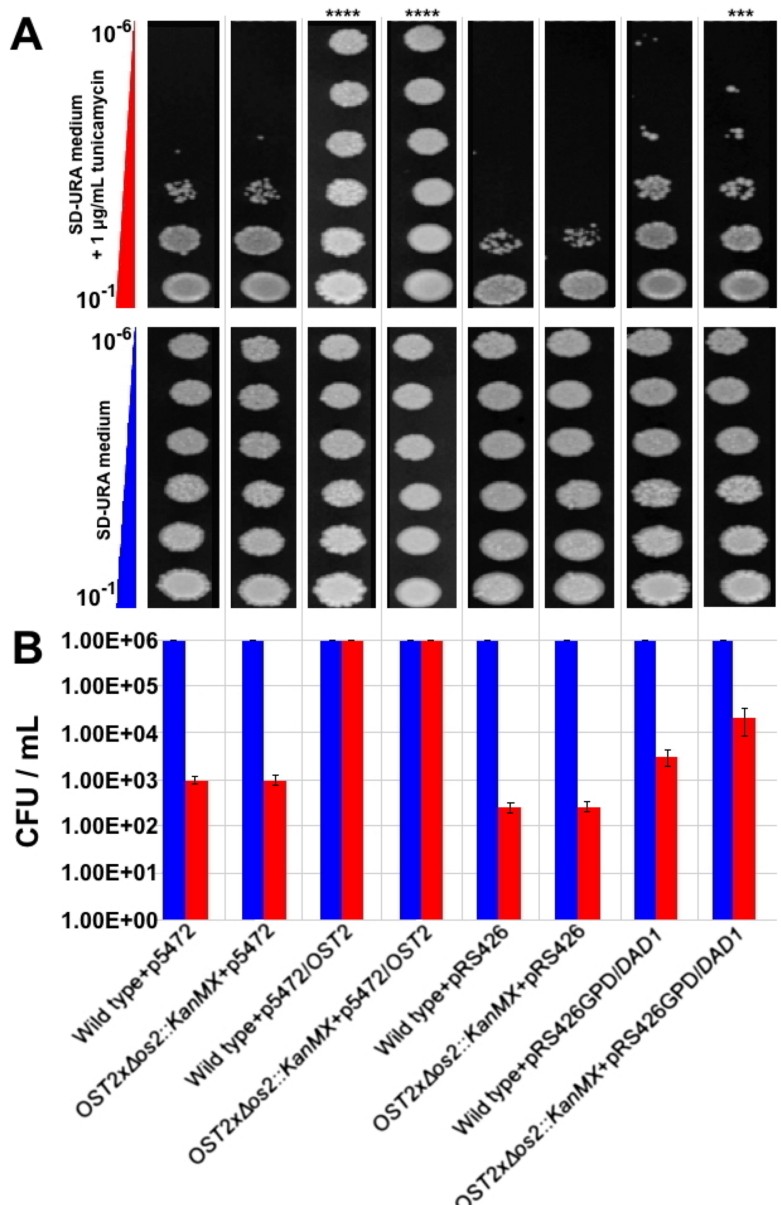

**Figure 5  Complementation of the yeast *ost2* knockout strain with the *B. braunii DAD1* cDNA.** Serial dilutions ($10^{-1}$ –$10^{-6}$) of yeast cells were plated in SD—URA medium containing 1 mg mL$^{-1}$ tunicamycin and incubated at 28 °C for 48 h (A). Viability of wild type and Δ *ost2* (*OST2x* Δ *ost2::KanMX*) was calculated after counting number of colonies in the presence or absence of tunicamycin (B). Mean ± SE, $n = 3$.

*Rao et al., 2007*; *Ambati, Ravi & Aswathanarayana, 2010*; *Hifney & Abdel-Basset, 2014*). Moreover, those studies focused on hydrocarbon production and showed results over many days of the algal growth cycle, which is a much longer time period in comparison to our studies. NaHCO$_3$ has previously been studied in *B. braunii* as a carbon source in comparison to glucose, sucrose, and sodium acetate, and showed no effect on the *B. braunii* growth rate after 15 days of growth with NaHCO$_3$ (*Kong et al., 2012*). No effect was observed on the *B. braunii* growth rate after 15 days of growth with NaHCO$_3$. However,

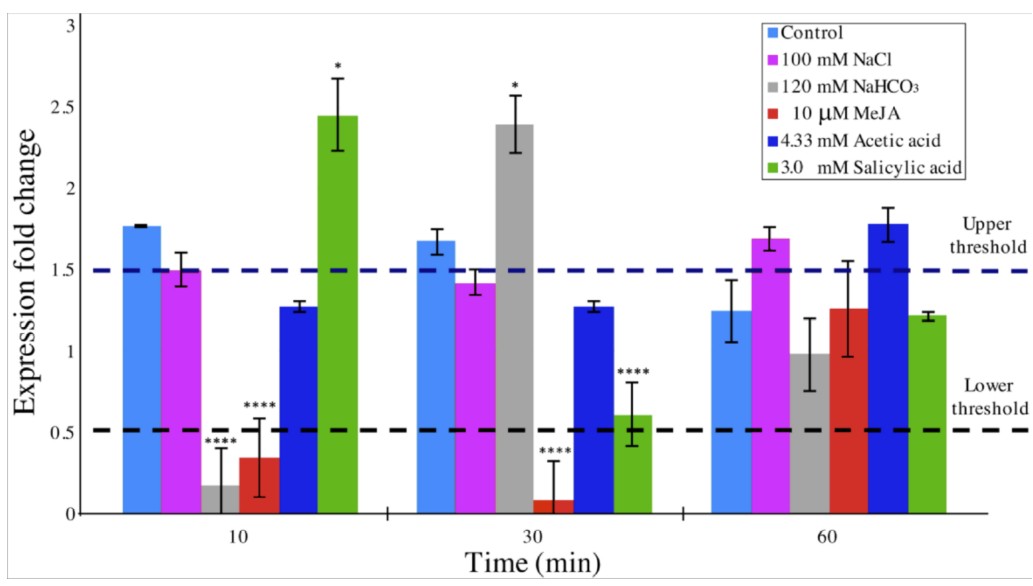

**Figure 6** **Relative transcript abundance of *B. braunii DAD1* gene during stress treatments.** qRT-PCR experiments were carried out and data were analyzed by the $2^{-\Delta\Delta CT}$ method. Gene expression data for all samples were normalized to *β-tubulin* gene expression and start time of each treatment. The dashed lines indicate the lower and upper threshold for relative regulation. Significance, indicated by ****$p < 0.0001$, ***$p < 0.001$, **$p < 0.01$ and $p < 0.05$ is relative to the control and was calculated by 2way ANOVA Multiple comparisons ($\alpha < 0.05$), $n = 2$.

NaHCO$_3$ increased carotenoid production in *B. braunii* although this increase was lower in comparison to the carotenoid production seen in response to NaCl and high light intensity in 21-day-old cultures (*Ambati, Ravi & Aswathanarayana, 2010*) or with CO$_2$ addition (*Tenaud, Ohmori & Miyachi, 1989*; *Wang et al., 2003*; *Tanoi, Kawachi & Watanabe, 2011*; *Hifney & Abdel-Basset, 2014*). Stress by acetic acid has not been studied in *B. braunii*, but it has been shown to induce PCD in other algal systems. A burst of hydrogen peroxide was reported after a 10 min exposure to acetic acid in *C. reinhardtii*, followed by PCD and abundant release of volatile organic compounds (VOC), which were proposed to act as chemical signals to other cells (*Zuo et al., 2012a*). The stress effect was due to the acetic acid instead of low pH because PCD was not induced when the pH was set at 4.0 with HCl (*Zuo et al., 2012b*). However, when the pH was set at 4.0 with HCl, no effect was reported (*Zuo et al., 2012b*). In our case, acetic acid at 4.33 mM was the second highest ROS inducer in *B. braunii* after 60 min of treatments, followed by a decrease in the later time points (Fig. 2F).

Although the effect of SA and MeJA in plants have been widely investigated as inducers of ROS during defense mechanisms against pathogen attack (*Oliveira, Varanda & Félix, 2016*; *Sewelam, Kazan & Schenk, 2016*), their role in algal metabolism is not yet clear. Studies with algae have shown that treatment of *Chlorella vulgaris* with SA induced salinity tolerance by increasing the internal osmotic pressure after production of high levels of carotenoids, soluble carbohydrates, and soluble proteins after seven days of culture (*Ismail et al., 2011*). Also, in *Tetraselmis suecica* an increase in carotenoid production was observed after culturing 7 days with SA, but this was not seen for *Dunaliella salina* (*Ahmed et al., 2015*). With MeJA, *D. salina* showed a significant increase in carotenoid production, but this

was not seen in *T. suecica* (*Ahmed et al., 2015*). In the microalga *Schizochytrium mangrovei*, the concentration of squalene, a natural antioxidant, increased significantly after 3 h of MeJA treatment (*Yue & Jiang, 2009*). Our data shows that *B. braunii* does indeed produce ROS in response to SA and MeJA treatment. A 3 mM SA treatment strongly induced ROS production after 60 min with no significant change until 120 min (Fig. 2D). Additionally, 10 $\mu$M MeJA increased ROS production after 60 min, which decreased in later time points (Fig. 2E).

## Hydrocarbon and biomass recovery after stress induction

In this study we did not make a detailed analysis of the hydrocarbon composition, but interestingly the main VOCs induced during PCD in *C. reinhardtii* after acetic acid treatment were mainly hydrocarbon derivatives like 3-methyl-2-pentanone, hexanal and 4-methyl-3-hexanone (*Zuo et al., 2012b*). The observed effect of acetic acid on *B. braunii* hydrocarbon production may be due to its availability for conversion into pyruvate, a precursor for botryococcene production through the MEP isoprenoid pathway (*Metzger & Largeau, 2005*). Previously, it was reported that three-week-old *B. braunii* cultures treated with $NaHCO_3$ to increase $CO_2$ concentrations showed a 1:1–1:3 increase in biomass and hydrocarbon production (*Dayananda et al., 2007*). The effect of salinity on this Showa strain of *B. braunii* by NaCl treatment was described as harmful for hydrocarbon production and biomass accumulation (*Yoshimura, Okada & Honda, 2013*). Same authors mention that the best conditions for growing *B. braunii* are at 30 °C with 0.2–5% $CO_2$ where the fastest grow rate is obtained (0.5 day$^{-1}$ which means a doubling time of 1.4 days), under these conditions, the hydrocarbon productivity also increases.

## Identification of *B. braunii* genes related to stress conditions

It is possible that the inoculation into fresh culture medium was sufficient to stress the cells and induce the expression of the *FER2*, *MBF1A* and *DAD1* genes. This could suggest that the increased amount of DAD1 protein may help in the recovery and growth of the cells after inoculation until they are fully recovered after 20 days. This hypothesis has some support from other studies. It has been reported that the *DAD1* gene was up-regulated in callus cells of *Agapanthus praecox* during the recovery stage following inoculation into fresh medium after cryopreservation (*Zhang et al., 2015*). It was important to know the expression change of these genes (Fig. 3) because with the low growing rate of 1.4 days doubling time, *B. braunii* cells required around 18 days to recover from the dilution effect after inoculation. Once recovered, we collected the cells, washed them and after 72 h they were subjected to the different stress treatments. In these conditions we were able to see the stress response of the *DAD1* gene in min without the effect of dilution which was higher after 7–8 days.

## Isolation of the *B. braunii DAD1* cDNA and analysis of the corresponding protein

Sequence comparison among DAD1 homologues suggests a conserved role in plant and animal PCD events (*Van der Kop et al., 2003*). However, there are interesting differences at the amino end of the DAD1 proteins where the highest sequence variability is seen for

the algal DAD1 proteins compared to the other DAD1 proteins shown in Fig. 4B. The DAD1 N-terminus may be important for its association with other proteins. It is known that the mammalian OST complex is composed of three membrane proteins: ribophorin I (RI), ribophorin II (RII), and OST48 all bound to the endoplasmic reticulum, and DAD1 has been proposed as a fourth subunit of this complex (*Silberstein, Kelleher & Gilmore, 1992*). Yeast two-hybrid experiments revealed affinity between the N-terminal region of cytoplasmic mammalian DAD1 and the cytoplasmic tail of OST48, which may fix DAD1 firmly into the OST complex (*Fu, Ren & Kreibich, 1997*). Up to now it is unknown if algae have a similar complex, although *DAD1* has been involved in PCD regulation. It has been reported that *DAD1* gene expression is high in metabolically active healthy plant tissues, but declines in the organs committed to die either by senescence or other stress conditions (*Orzáez & Granell, 1997*).

## Complementation of the yeast *ost2* knockout strain with the *B. braunii DAD1* cDNA

Expression of the *B. braunii DAD1* cDNA in wild-type yeast or the *ost2* knockout line showed a slight but significant increase in resistance to tunicamycin (Figs. 5A & 5B). This data suggests the *B. braunii DAD1* cDNA is capable of complementing the function of the yeast *OST2* gene and is capable of functioning in the glycosylation of proteins. Complementation of the *ost2* knockout with the *B. braunii DAD1* was not as strong in the presence of tunicamycin compared to expression of the *OST2* gene in the *ost2* knockout (Fig. 5B). This could be due to the promoters used or the differences in codon usage between the native *OST2* and the heterologous *DAD1*. Expression of the *OST2* gene in these experiments was controlled by the native promoter and terminator sequences, which should allow for the correct complementation in the *ost2* knockout. In comparison, the *B. braunii DAD1* cDNA was expressed from the constitutive GPD promoter.

## qRT-PCR of the *B. braunii DAD1* gene

Considering that relative quantification has been proposed as the best method of analyzing data for qRT-PCR at real time in comparison to absolute quantification, we decided to perform experiments of *DAD*1 gene expression under stress treatments relative to untreated control conditions in a time course study (*Livak & Schmittgen, 2001*). The $2^{-\Delta\Delta CT}$ algorithm requires at least one housekeeping gene for analysis. We used *β-tubulin* in our experiments which indeed showed a uniform gene expression in all samples (*Zhang, Ruschhaupt & Biczok, 2010*) (Fig. 6). With this method, significant fold change in *DAD*1 gene expression were observed for short times between 10 and 30 min under some treatments like SA, MeJA and NaHCO₃. At 60 min the equilibrium seems to be reestablished. These results suggest that *DAD1* may be a part of the stress response mechanism that helps to minimize effects produced by ROS accumulation under stress treatments. Also, these results agree with the proposed role of *DAD1* gene in cell viability as previously reported for *C. reinhardtii* (*Pérez-Martín et al., 2014*). On the other hand, down regulation of *DAD1* expression in *C. reinhardtii* has been observed in response to UV light, but only after longer times between 30 min and 6 h (*Moharikar, D'Souza & Rao, 2007*). The

short times of the responses reported in this work suggests that *B. braunii* quickly senses the stress inducers and produces ROS. Additionally, the expression changes in the *DAD1* gene are manifested in min.

## CONCLUSIONS

*B. braunii* has been cultured under a large variety of culture conditions including temperature variations (*Kalacheva et al., 2002*), nitrogen limitation (*Zhila, Kalacheva & Volova, 2005*); salinity differences (*Rao et al., 2007*; *Zhila, Kalacheva & Volova, 2011*), variable light, nutrients, and cultivation time (*Ruangsomboon, 2012*), different sugars as carbon sources (*Weetall, 1985*), different mineral salts like potassium nitrate, magnesium sulfate, dihydrogen potassium phosphate and ferric citrate (*Dayananda et al., 2005*), and $CO_2$ levels (*Yoshimura, Okada & Honda, 2013*). However, most of these studies have been done over long time periods of days in order to find the best condition to increase hydrocarbon production. In comparison very little is known about the timing of stress responses in this organism. Using fluorescent staining to identify ROS production after a stress response (Fig. 1), our results show that *B. braunii* ROS production was triggered at short times (10 min) after treatment with all stress inducers. Biomass productivity was not affected by any of the stress inducers and hydrocarbon production was increased only by acetic acid (Table 2).

The DAD1 protein is a highly conserved suppressor of PCD (*Sinha et al., 2015*) and although it is not exactly clear how this protein is related to PCD control, it protects plant and algal cells from this lethal process (*Lindholm et al., 2000*; *Yamada et al., 2004*; *Moharikar, D'Souza & Rao, 2007*). DAD1 inhibits PCD during several plant processes like senescence, UV-C exposure, and seed development (*Rantong & Gunawardena, 2015*). According to the transcriptome data, we found that the *DAD1* gene of *B. braunii* was highly expressed after inoculation of cells into fresh medium and declined after 15 days (Fig. 3). During stress treatments, *DAD1* was strongly up-regulated by SA in the first 10 min (Fig. 6). These results suggest that *DAD1* may help minimize stress effects induced by the ROS induced by SA treatment. Even though *B. braunii* cells are embedded in a complex ECM, externally applied molecules are still able to penetrate the cells and induce ROS production (Fig. 1), within minutes resulting in stronger transcription of the gene *DAD1* (Fig. 6).

Finally, recovery of the yeast *OST2* phenotype by the *DAD1* cDNA of *B. braunii* suggests that the corresponding protein is involved in the N-glycosylation process as is expected for the function of the DAD1 protein. It is possible that *DAD1* may be part of a rapid sensing or defense mechanism allowing the alga to adapt as fast as possible to changing environmental conditions. In spite of these results, we cannot directly correlate DAD1 function in the PCD process of *B. braunii*. More experiments are required to confirm the role of *B. braunii* DAD1 in PCD, such as those carried out in *A. thaliana* where *DAD1* overexpression clearly demonstrated PCD inhibition under stress conditions (*Hogg et al., 2011*).

## ACKNOWLEDGEMENTS

We thank Dr. Plinio Guzmán-Villate from Departamento de Ingeniería Genética CINVESTAV-IPN Irapuato, Guanajuato, México for his expertise, valuable advice, and support with the yeast work.

### Funding

This work was supported by a PhD scholarship to IC-C from Consejo Nacional de Ciencia y Tecnología (CONACYT) Mexico, a grant from the 2012 Texas A&M University—CONACYT Collaborative Research Grant Program to EL-G and TPD, and grant # 1240478 from the National Science Foundation (NSF) Emerging Frontiers in Research and Innovation (EFRI) to TPD. The funders had no role in study design, data collection and analysis, decision to publish, or preparation of the manuscript.

### Grant Disclosures

The following grant information was disclosed by the authors:
Consejo Nacional de Ciencia y Tecnología (CONACYT) Mexico.
2012 Texas A&M University—CONACYT Collaborative Research Grant Program.
National Science Foundation (NSF) Emerging Frontiers in Research and Innovation (EFRI): #1240478.

### Competing Interests

The authors declare there are no competing interests.

### Author Contributions

- Ivette Cornejo-Corona and Daniel R. Browne conceived and designed the experiments, performed the experiments, analyzed the data, wrote the paper, prepared figures and/or tables, reviewed drafts of the paper.
- Hem R. Thapa conceived and designed the experiments, performed the experiments, analyzed the data, wrote the paper, reviewed drafts of the paper.
- Timothy P. Devarenne conceived and designed the experiments, analyzed the data, contributed reagents/materials/analysis tools, wrote the paper, reviewed drafts of the paper.
- Edmundo Lozoya-Gloria conceived and designed the experiments, analyzed the data, contributed reagents/materials/analysis tools, wrote the paper, prepared figures and/or tables, reviewed drafts of the paper.

### DNA Deposition

The following information was supplied regarding the deposition of DNA sequences:
GenBank: KU746841.

### Data Availability

The raw data have been supplied as Supplementary Files.

## Supplemental Information

Supplemental information for this article can be found online at http://dx.doi.org/10.7717/peerj.2748#supplemental-information.

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
