# Peer review of "Stress responses of the oil-producing green microalga Botryococcus braunii Race B"

_PeerJ, doi:10.7717/peerj.2748_

## Round 0.1 · original submission · Major Revisions

Please follow the advice from both reviewers and provide results and discussion in separate sections. In addition, please rework especially the sections on PCD and the RNA-seq experiment.

Reviewer 1 ·

Basic reporting

The structure seems appropriate.The manuscript is clearly written.
The manuscript includes sufficient Introduction, but I would say, it is too lengthy. Relevant literature is referenced, with the correct notion on little evidence on the role of ROS in PCD or PCD-like processes in microalgae.
Results and Discussion are combined, I think these sections should be separated to clarify and appropriately discuss the findings of the work.
Raw data are available.

Experimental design

This is original and primary research.
The research question is identified .
The manuscript starts with identification of conditions which lead to induction of ROS production. To this aim, the authors tested several chemicals that have been reported to induce ROS and PCD-like effects in some publications on algae.
Staining with CellROX Green reagent was used to visualize ROS production by quantification of stained niclei. Quantification of these results by the number of positive colonies is presented.
line 289 and Figure 1: micrographs seem all different to me: why cells are colored yellow on white light micrographs? ?
There are several points to note: i) whether the treatment at 36oC for 30 min affected staining with CellROX Green reagent in control cells? This is by itself seems to be drastic treatment; 2) in almost all cases the response to "inducer" was not dose-dependent? 3) in my opinion, the short –time exposure for ~10 minutes in all cases was sufficient to visualize the effect, this notion is reflected by RT-PCR results on relative expression of DAD1 gene. These results are also not very conclusive. I would recommend performing qPCR within the short time span.

Validity of the findings

The data are statistically treated.
The sequence of DAD1 is deposited at NCBI, a link to transcriptomics data is provided.

There was no correlation found between growth rate, hydrocarbon productivity and exposure to the stressors, with the exception of the treatment eithacetic acid, the most drastic treatment, and not sure -- the physiologically relevant treatment.

DAD1 gene was highly expressed in control culture cultured during 18 days; a similar inoculation mode also applied to other experimental treatments which in turn might have affected the gene expression levels. Please elaborate.

Additional comments

The manuscript can be appropriately divided into Results and Discussion sections .
The part on DAD1 identification and characterization is most interesting, and seems to be a stand-alone manuscript.
I think that a stronger support is needed to argue for the DAD1 involvement in PCD in this alga.
I recommend major revisions to this manuscript.

Reviewer 2 ·

Basic reporting

-Clear, unambiguous, professional English language used throughout : YES

-Intro & background to show context: YES

-Literature well referenced & relevant : YES

-Structure conforms to PeerJ standard, discipline norm, or improved for clarity: YES though results and discussion are not in separate sections

-Figures are relevant, high quality, well labelled & described: not all (see details)

-Raw data supplied (See PeerJ policy): YES

Experimental design

Original primary research within Scope of the journal: YES

Research question well defined, relevant & meaningful. It is stated how research
fills an identified knowledge gap: YES

Rigorous investigation performed to a high technical & ethical standard: YES, except figure 6

Methods described with sufficient detail & information to replicate : YES, except maybe RNA-seq (see details)

Validity of the findings

Data is robust, statistically sound, & Controlled: YES, except figure 6 (see details)

Conclusion well stated, linked to original research question & limited to supporting
results: YES

Additional comments

Detailed comments
1. RNA-seq (Lines 180-183, and figure 3): it is not clear if RNA-seq data used to make figure3 has been published elsewhere by the Chappell lab. If data is not yet available, it is not possible to assess quality of RNA-seq.
2. RNA-seq (Line 180-183): was race B used for RNA-seq?
3. RNA-seq: I am not sure figure 3 is so useful because it deals with expression of selected gene under non-stress conditions (growth curve). DAD1 could have been selected without info of figure 3. Maybe figure 3 could me moved to supplemental.
4. RT-PCR (Figure 6): signal for housekeeping gene seems saturated under many conditions and for time points, which casts doubt on validity of results presented in figures 6B-F.
5. Images for timepoints 1 min and 120 min are missing on figure 6A.

---

## Round 0.2 · accepted · Accept

All points raised by the two reviewers were carefully addressed throughout the manuscript. Thank you very much.